# Correlation between Gut Microbiota and Six Facets of Neuroticism in Korean Adults

**DOI:** 10.3390/jpm11121246

**Published:** 2021-11-24

**Authors:** Eunkyo Park, Kyung Eun Yun, Mi-Hyun Kim, Jimin Kim, Yoosoo Chang, Seungho Ryu, Hyung-Lae Kim, Han-Na Kim, Sung-Chul Jung

**Affiliations:** 1Department of Biochemistry, College of Medicine, Ewha Womans University, Seoul 07804, Korea; eunkyo85@gmail.com (E.P.); hyung@ewha.ac.kr (H.-L.K.); 2Center for Cohort Studies, Total Healthcare Center, Kangbuk Samsung Hospital, School of Medicine, Sungkyunkwan University, Seoul 04514, Korea; eun3579@hanmail.net (K.E.Y.); mh0303.kim@samsung.com (M.-H.K.); jimin.kim@samsung.com (J.K.); yoosoo.chang@gmail.com (Y.C.); sh703.yoo@gmail.com (S.R.); 3Department of Occupational and Environmental Medicine, Kangbuk Samsung Hospital, School of Medicine, Sungkyunkwan University, Seoul 03181, Korea; 4Department of Clinical Research Design and Evaluation, SAIHST, Sungkyunkwan University, Seoul 06355, Korea; 5Medical Research Institute, Kangbuk Samsung Hospital, School of Medicine, Sungkyunkwan University, Seoul 03181, Korea; 6Graduate Program in System Health Science and Engineering, Ewha Womans University, Seoul 07804, Korea

**Keywords:** microbiota, neuroticism, gut–brain axis, 16s sequencing, anxiety, depression, vulnerability

## Abstract

A person high in neuroticism is more likely to experience anxiety, stress, worry, fear, anger, and depression. Previous studies have shown that the gut microbiota can influence personality and mental disorders, including stress, anxiety, and depression, through the gut–brain axis. Here, we investigated the correlations between the sub-facet of neuroticism and gut microbiota using the Revised NEO Personality Inventory and the 16S rRNA gene sequencing data 784 adults. We found that the high anxiety and vulnerability group showed significantly lower richness in microbial diversity than a group with low anxiety and vulnerability. In beta diversity, there was a significant difference between the low and high groups of anxiety, self-consciousness, impulsiveness, and vulnerability. In taxonomic compositions, Haemophilus belonging to Gammaproteobacteria was correlated with the Neuroticism domain as well as N1 anxiety and N6 vulnerability facets. The high N1 anxiety and N6 vulnerability group was correlated with a low abundance of Christensenellaceae belonging to Firmicutes Clostridia. High N4 self-consciousness was correlated with a low abundance of Alistipes and Sudoligranulum. N5 impulsiveness was correlated with a low abundance of Oscillospirales. Our findings will contribute to uncovering the potential link between the gut microbiota and neuroticism, and the elucidation of the correlations of the microbiome–gut–brain axis with behavioral changes and psychiatric cases in the general population.

## 1. Introduction

Personality traits represent the different behavioral, emotional, and cognitive patterns of individuals. Specific personality traits have been suggested to affect the risk of physical illnesses and behavior-related health risk factors [1,2]. In previous studies, interest in personality traits, particularly neuroticism, has been motivated by the fact that personality traits remain stable into adulthood [3], have genetic–environmental underpinnings [4], and are also predictive of late-life developments such as cognitive dysfunction [5] and psychiatric symptoms [6]. Neuroticism can be viewed as a heterogeneous trait consisting of multiple facets, including anxiety (N1), hostility (N2), depression (N3), self-consciousness (N4), impulsiveness (N5), and vulnerability to stress (N6) [7]. These facets are highly correlated but partially distinct, including anger, sadness, anxiety, worry, and hostility [8]. However, most studies have used measures that capture only a domain level of relevant personality traits. They rarely assessed these facets. These more circumscribed facets have greater predictive power for specific behavioral and health outcomes than the broader domain-level neuroticism [9,10].

Recently, the importance of the gut microbiome in human health has been in the spotlight [11]. Since the gut microbiota play a central role in the gut–brain axis that regulates mood and behavior, they can affect various aspects of normal psychology, such as emotion, cognition, stress management, and social behavior in addition to physical health [6,12]. The gut microbiota are also correlated with the predisposition of personality and mental disorders [13,14]. Dysbiosis in the gut microbiota may increase pro-inflammatory communication, which in turn increases intestinal permeability, which can lead to an inflammatory response to stress systems of the brain either directly or vagus/visceral afferent [11,12]. Pro-inflammatory communication has been shown to impair negative feedback within the hypothalamic–pituitary–adrenal (HPA) axis and induce hypercortisolemia [11]. Elevated cortisol levels and inflammatory markers are reported to be associated with anxiety and depressive disorders [12]. This bidirectional communication implicates that increased cortisol delivered to the body can affect immune function, intestinal permeability, and gut microbiota [12].

In a previous study, we have reported the correlations between the gut microbiome and personality traits. Highly neurotic individuals were likely to show a high abundance of Gammaproteobacteria. However, the subscales of neuroticism were not considered in that study. Recent research has suggested that compositions of the human microbiome are linked to stress, depression, and broad personality traits [14,15,16,17]. Chronic stress can increase the risk of developing many psychiatric disorders such as anxiety, depression, and post-traumatic stress disorder [18,19]. Although the mechanisms underlying vulnerability to stress remain unclear, mounting evidence suggests that increased central inflammatory processes might be involved [20,21,22]. Indeed, the gut microbiome contributes to the depression-like behaviors and inflammatory processes in the ventral hippocampus of stress vulnerable rats [15]. Although the Revised NEO Personality Inventory (NEO-PI-R) contains the six facets that are the subscales of neuroticism, there are no reported studies on the correlation between these neuroticism facets and the gut metagenome. Neuroticism is hierarchically defined by the specific facets, which can provide a more in-depth description of the correlation between personality and gut microbiota. Moreover, as a more circumscribed facet, the trait of neuroticism measures a narrower phenotype, which can increase the statistical power by reducing phenotyping variability [23].

This study extends our previous study by examining the microbial diversity and taxonomic composition of gut microbiota in a large population-based sample while utilizing well-validated and comprehensive measures of personality that captures both a domain and specific facets of neuroticism.

## 2. Materials and Methods

### 2.1. Subjects

In this study, participants were enrolled in the Kangbuk Samsung Study for Korean men and women who undergo a comprehensive test every year or every two years at the health examination center at Kangbuk Samsung hospital in South Korea. From June 2014 to September 2014, 1463 adult participants aged 23 to 78 who received comprehensive health checkups (907 males and 556 females) gave fecal samples, of which 784 participants also completed a personality questionnaire. All participants who met any of the exclusion criteria as described were not enrolled in this study (Figure 1). Only those with complete personality tests were enrolled. Those who used antibiotics within 6 weeks prior to enrollment or cholesterol-lowering medications or probiotics within 4 weeks prior to enrollment were excluded. Additionally, participants who were diagnosed with mental diseases, such as depression or panic disorders, were excluded. A total of 90 samples with under 5000 sequences per sample were also excluded. Finally, a total of 784 subjects (489 males and 295 females) were enrolled in this study.

The research protocol was approved by the Institutional Review Board of Kangbuk Samsung Hospital (approval number: 2013-01-245-12). After explaining the nature of the study and the possible consequences, participants provided written informed consent. All applicable institutional and governmental regulations concerning the ethical use of human volunteers were followed during this research.

### 2.2. Fecal Sample Collection and 16S rRNA Gene Compositional Analysis

Fecal samples were immediately frozen at −20 °C after defecation. They were stored at −70 °C within 24 h. Within 1 month, DNA was extracted from fecal samples using a fecal DNA extraction kit (MO BIO Laboratories, Carlsbad, CA, USA) according to the manufacturer’s instructions. Amplification and sequencing were performed to analyze bacterial communities, as described previously [13]. Genomic DNA was amplified using fusion primers targeting the variable V3 and V4 regions of the 16S rRNA gene with indexing barcodes. Samples were pooled for sequencing on an Illumina MiSeq platform (Illumina, San Diego, CA, USA) according to the manufacturer’s instructions [24,25].

The DADA2 plugin of the QIIME2 package (version 2019.7, https://qiime2.org (accessed on 16 February 2021)) [26] was used to perform sequence quality control, such as filtering low-quality sequences and chimeras, and to construct a feature table of amplicon sequence variants (ASVs). ASVs were generated by denoising with DADA2 and regarded as 100% operational taxonomic units (OTUs). To identify amplicon sequence variants (ASVs) from non-chimeric sequences, an open-reference ASV picking approach was performed using representative sequences with pre-assigned taxonomy from SILVA DB (version 138). This analysis was performed in QIIME2 (version, 2020. 2), with a 99% similarity threshold [27,28]. Contingency-based filtering was used to filter features from a table contingent on the number of samples in which they were observed. We filtered features that were present in only one sample based on the assumption that these features were not due to real biological diversities but due to polymerase chain reaction (PCR) or sequencing errors such as PCR chimeras.

### 2.3. Personality Assessment

Personality traits were assessed using the Korean version of the Revised NEO Personality Inventory (NEO PI-R), which is a 240-item measure of the five factors of personality (PSI Consulting Corp., Seoul, Korea) [7]. The Korean version of the NEP PI-R has been used in the Korean population with good reliability and validity [29]. The NEO PI-R consists of 30 facets, six for each dimension of the five personality traits. These facets of neuroticism are anxiety (N1), angry hostility (N2), depression (N3), self-consciousness (N4), impulsivity (N5), and vulnerability (N6). Raw scores of each dimension were combined. For qualitative analysis, we divided the subjects into two groups according to raw score of the subfacet of neuroticism. The low group and the high group were defined by 25th quartiles, with the low group being in the first quartile (≤25th percentile) and the high group in the fourth quartile (≥75th percentile) based on neuroticism facet raw scores. We categorized these two groups for the facets of neuroticism separately for men and women because of the basic differences in the personality scores by gender. As a result of the basic differences between male and female scores, we separated males and females and set the scores for each. In a previous study [13], the raw score of each dimension was converted to T-scores, which were calculated sex-separately with the normative Korean data. In this study, we decided to use the raw score instead of converted score. To avoid bias caused by sex differences, males and females were grouped separately.

### 2.4. Statistical Analysis

All basic statistical analyses were performed with PSPP version 1.4.0. The feature table was rarefied to 5011 sequences per sample by random sampling in QIIME2 before diversity analysis. Alpha diversity measures of richness, community diversity, evenness, and phylogenetic diversity of gut microbial taxa were presented as observed features, Shannon index, Pielou’s evenness, and Faith’s phylogenetic diversity (PD), respectively. For measuring beta diversity, Bray–Curtis, Jaccard, and unweighted and weighted UniFrac values were calculated to determine the dissimilarity between groups [30,31,32] The *p* values were calculated using Kruskal–Wallis test. Differences in beta diversity between the groups were compared using pairwise permutational multivariate analysis of variance (PERMANOVA with 999 permutations).

For composition analysis, Analysis of Composition of Microbiome (ANCOM) [14] was used to compare the log-ratio of different abundances of gut microbial taxa in the neuroticism facet groups. ANCOM compares the relative abundance of taxa among multiple groups by the log-ratio of the abundance of each taxon to the abundance of all the remaining taxa one at a time. To adjust for confounding variables (age, sex, and BMI), we used the ANCOM2 code shared by the author from the original ANCOM paper, which could deal with covariates. Correlation and comparison between the abundance of taxa and the six facets of neuroticism were calculated using the Multivariate Association with Linear Models (MaAsLin) using an online protocol package (http://huttenhower.sph.harvard.edu/galaxy (accessed on 4 Feburary 2021)) [33,34]. Analyses included covariate adjustments for age and BMI, which could affect both gut microbiome composition and personality. Additionally, microbial community function was evaluated by predictive metagenome (microbial DNA) analysis using PICRUST2. PICRIST2 is a developed phylogeny-based computational tool that can predict the functional capacity of microbial communities by correlating the species present to reference databases of microbial genomes. We performed PICRUST2 according to the protocol (https://github.com/picrust/picrust2/wiki/q2-picrust2-Tutorial (accessed on 15 March 2021)) [35]. DADA2 variants were normalized using the 16S rRNA copy number, and KEGG values (Kyoto Encyclopedia of Genes and Genomes) were predicted. Results that aggregated to level three of the KEGG analysis module were further explored with STAMP [36].

## 3. Results

### 3.1. Baseline Characteristics of the Subjects

Descriptive statistics for the study participants are shown in Table 1. A total of 15,055,235 high-quality paired sequences were obtained from the 784 samples (mean age, 43.9 years), with a mean of 23,669 feature counts. Among the 784 subjects, 489 (62.37%) were men and 295 (37.63%) were women.

The low group and the high group were defined by 25th quartiles, with the low group being in the first quartile (≤25th percentile) and the high group being in the fourth quartile (≥75th percentile) of neuroticism facet raw scores. For age, the higher the score of N1 anxiety, N3 depression, and N5 impulsiveness, the younger the age. In the case of N5 impulsiveness, there were significant differences between groups not only in age and BMI but also in nutrient factors. Moreover, the high groups of the N neuroticism domain, N5 impulsiveness, and N6 vulnerability showed significantly higher total energy intake.

### 3.2. Comparison of Biodiversity between Low- and High-Scored Groups of Facets of Neuroticism Facets

The mean depth of sequences was 23,669 per sample and the number of features was 3524 in the 784 subjects. After rarefying the feature tables to 5011 sequences per sample, the neuroticism domain exhibited different alpha diversity in Faith’s PD between the low and high groups (Figure 2). The low group of N1 anxiety showed greater diversity in observed features, Faith’s PD, and Shannon’s diversity. The high group of N6 vulnerability showed lower alpha diversity in observed features, Faith PD, and Shannon’s diversity.

Beta diversity analysis indicates the extent of similarities and differences among microbial communities. To quantify beta diversity, both phylogenetic and non-phylogenetic methods were used with unweighted and weighted UniFrac and Jaccard distances and Bray–Curtis dissimilarity, respectively (Figure 3). For the neuroticism domain, we confirmed significant differences between the low and high groups in unweighted Unifrac distance (*p* < 0.05, PERMANOVA), similar to the results of our previous study [13]. For facets of neuroticism, there were significant differences in unweighted distances between the high and low groups of the N1 anxiety, N4 self-consciousness, N5 impulsiveness, and N6 vulnerability facets, respectively. For N2 and N3, we could not find statistical differences in the structure of the gut microbial community between the low and high groups (Appendix A).

### 3.3. Correlations of Taxonomic Composition with Six Facets of Neuroticism

To better understand how the microbial taxonomic compositions changed with the six facets of neuroticism, we compared the relative abundance levels of low and high groups in each facet of neuroticism. To control for covariates, we controlled for age, sex, and BMI to determine if there was a significant association between microbial taxa and the facets of neuroticism. Based on W statistics by ANCOM, we identified 13 taxa associated with the six facets of neuroticism from phylum to species level after adjusting for age, sex, and BMI (Table 2).

Pairwise comparisons were undertaken to identify the significant differences between the two groups (Figure 4). Beta diversity results were verified by pairwise ANCOM analysis. Indicated specific taxa were significantly different between the low and high groups. The neuroticism domain was positively associated with the family Pasteurellaceae (W = 79) as well as its upper taxa Pasteurellales (W = 43). The “W = 79” of Pasteurellaceae in the neuroticism domain indicated that this family was significantly different relative to 79 other families between the two groups. For N1 anxiety, the genus Haemophilus (W =2 31), including its family Pasteurellaceae (W = 77) and order Pasteurellales (W = 43), also showed positive associations with the neuroticism domain, while genus Christensenellaceae R.7 group (W = 221), including its family Christensenellaceae (W = 79) and order Christensenellales (W = 41), showed negative associations between the two groups. Genus Alistipes (W = 204) and its family Rikenellaceae (W = 71) and genus Sudoligranulum (W = 215) showed significantly lower abundance levels in the high N4 self-consciousness group. N5 impulsiveness was correlated with the order Christensenellales (W = 39) of Firmicutes, the family Marinifilaceae of Bacteroidota (W = 76), genus UCG.002 (W = 216), and *Clostridia UCG.014* (W = 73), including its order Clostridia UCG.014 (W = 44). The high group of N6 vulnerability was negatively correlated with genus Christensenellaceae *R.7 group* (W = 215) and its family Christensenellaceae (W = 68) and positively correlated with the genus Haemophilus (W = 218), its family Pasteurellaceae (W = 77), and its order Pasteurellales (W = 42) compared to the low group. However, results of N2 hostility and N3 depression did not show significantly different taxa between the low and high groups.

We also used linear discriminant analysis (LDA) of effect size (LEfSe) to determine the taxa that most likely explained the differences between the low and high groups. When performing the LEfSe analysis, we compared taxa not only on the basis of statistical significance but also based on the biological consistency of results and effect relevance. Figure 5 shows LEfSe results (LDA score > 3, *p* < 0.05) from the phylum level to genus level, which confirmed that the genus Haemophilus, family Pasteurellaceae, and order Pasurellales were significantly enriched in the group with a high neuroticism domain and its facets, except N2 hostility, in the high group compared with the low group (LDA > 3, *p* < 0.05).

### 3.4. Predicted Functional Metagenome in Personality Groups

Based on functional predictions using PICRUSt and STAMP, we tried to detect differences in the KEGG Ortholog composition between the low and high groups for the five personality traits, but no significant differences were found (data not shown).

## 4. Discussion

In this large longitudinal study, we investigated the association between the gut microbiome and facets of neuroticism. Phylogenetic and non-phylogenetic measures of alpha diversity for gut microbiota were lower in the high group of neuroticism facets. There were significant differences in the compositions of gut microbiota between the two groups in this study.

Previous studies have reported the relationship of gut microbiota with personality. The majority of these studies showed a decrease in the overall bacterial diversity in the highly neurotic personality group [13,14,16]. We observed that individuals with high N1 anxiety or N6 vulnerability personality types had significantly reduced gut microbial diversity and showed differences in microbial composition compared to the subjects with low neuroticism personality.

These data are in line with the growing body of evidence for a bidirectional relationship between the gut microbiome and mental health [37]. Similarly, human studies have concluded that the microbiome is involved in psychopathology behavior, such as in autism, depression, anxiety, obesity, and anorexia nervosa, through the gut–brain axis [38,39,40]. Bidirectional interactions between the central nervous system and gut microbiota are directly maintained by the limbic system, the hypothalamic–pituitary–adrenal axis [41], and the sympathetic nervous system [42], or via indirect mechanisms, such as neurotransmitters and immune and metabolic pathways [43,44,45,46]. Recent studies have suggested that anxiety states are associated with stool consistency and that anxiety status might be associated with differences in the compositions of the gut microbiome through the induction of dysbiosis [14,47,48,49,50]. In addition, animal studies have previously shown that stress can alter the abundances of various microbial taxa and reduce the diversity of the gut microbiota [15,48,51,52,53,54].

Recent cohort studies of major depressive disorder (MDD) and anxiety disorders have reported lower bacterial alpha diversities in these patient groups relative to controls, as well as a higher relative abundance of the proteobacteria [47]. Existing research has indicated that anxiety and depression shared gut microbiota alterations, including lower alpha diversity, and a higher abundance of proteobacteria and toxin-releasing genera relative to controls [47,55]. The autonomic and circulatory systems carry distress signals to the gut. Then, immune cells as messengers convey psychological stress to the gut via the bone marrow-mediated pathway [56]. The increased inflammation that frequently accompanies stress and depression can cause blooms of pathogenic bacteria that encourage dysbiosis [57]. Animal studies have demonstrated that stress could affect the diversity of gut bacteria [58,59].

Our previous study [13] showed that the genus Haemophilus was associated with high neuroticism. In this facet study, Haemophilus and its upper-class taxa Pasteurellaceae and Pasteurellales showed positive correlations with high neuroticism, especially for the facets N1 anxiety and N6 vulnerability, as well as the neuroticism domain. We showed that several taxa were associated with the N1 anxiety index. Christensenellaceae, belonging to Firmicutes Clostridia, was negatively associated with N1 anxiety. In accordance with these results, previous studies in children have reported a reduction in Oscillospira of Firmicutes. Rumminococcaceae has a positive association with good health. It has been reported that lower levels of Oscillospira are linked to inflammatory disease [39,60]. Similarly, fecal microbiota of patients suffering from generalized anxiety disorder show a much lower prevalence of Lachnospira, which belongs to Firmicutes Clostridia [47]. These genera could be relevant to mental health due to their documented production of short-chain fatty acid compounds [61]. Reduced short-chain fatty acid (SCFA) production in GAD patients with intestinal barrier dysfunction could compromise proper immune responses and ultimately contribute to brain dysfunction [62]. We found that the characteristic gut microbiota identified here were slightly different from those identified in previous studies. These distinctions might have arisen from differences in the sample size, demographic, and clinical characteristics of participants, individual differences, and the statistical approach applied to identify gut microbiota. A model of social disruption among adult mice has demonstrated that exposure to stress can result in substantial changes in gut microbiota composition [63]. Bailey et al. showed decreases in Bacteroides spp. and increases in Clostridium spp. abundance in stress-induced mice [64], while we found a lower abundance of Clostridia of Firmicutes and Bacterioidales in this study.

There was a significant difference in age between the two groups in the N1 anxiety, N5 impulsiveness, and N6 vulnerability facets. Although it was found that the group with higher anxiety, impulsiveness, and vulnerability to stress scores was significantly younger, the analysis was not stratified by age, but it was analyzed with age as a covariate. In addition, in the case of the N5 impulsiveness facet, we found that BMI and the intake of nutrient factor were significantly different between the two groups. Further studies on the association between dietary intake and impulsiveness are needed. Consistent with these previous studies, we found differences in gut microbiota according to facets of neuroticism index in adults. The 16S approach is well suited for the analysis of many samples. However, it offers limited taxonomical and functional resolution. Additional experiments are needed to understand the biochemical and functional consequences of these predictions.

## 5. Conclusions

This study was a follow-up study that revealed the relationship between personality and the metagenome, and we analyzed the sub-facets of neuroticism among the personality traits. Our findings demonstrated that gut microbiota dysbiosis has a close relationship with gastrointestinal and behavioral manifestations of neuroticism. This study will contribute to elucidating potential links between the gut microbiota and neurotic personality, yielding promising potential for developing and testing personality- and microbiota-based interventions for promoting mental health.

## Figures and Tables

**Figure 1 jpm-11-01246-f001:**
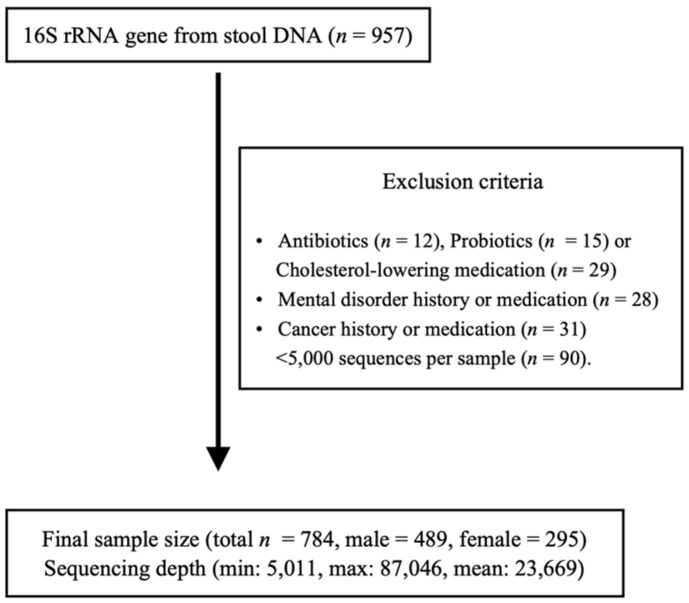
Enrollment of subjects. Inclusion criteria: 1463 Korean adults aged 23 to 78 years who agreed to participate in this study and provided stool samples at Kangbuk Samsung Hospital Healthcare Screening Center out of those who underwent annual or biennial examinations during the study period between June 2014 and September 2014. Some individuals met several exclusion criteria.

**Figure 2 jpm-11-01246-f002:**
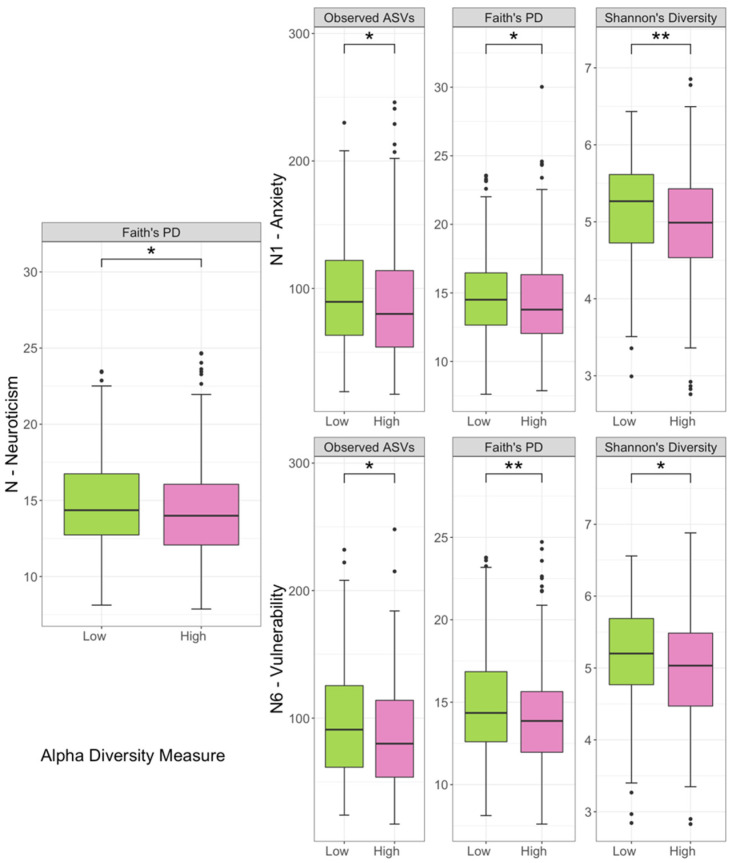
Alpha diversity among groups. Diversity was significant for phylogenetic diversity (Faith’s PD) in the neuroticism domain, observed features (ASVs), phylogenetic diversity, and Shannon index in N1 anxiety and N6 vulnerability. The *p* values were calculated using Kruskal–Wallis test. * *p* < 0.05, ** *p* < 0.01. Notched boxes indicate interquartile range (IQR) of 25th to 75th percentiles. The median value is shown as a line within the box. The notch indicates the 95% confidence interval for the median. Whiskers extend to the most extreme value within 1.5 × IQR. Possible outliers are shown as dots.

**Figure 3 jpm-11-01246-f003:**
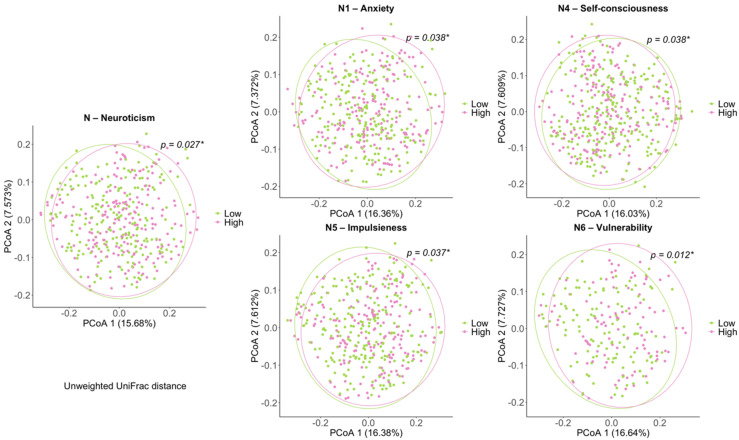
Principal Coordinate Analysis (PCoA) plots of beta diversity. Statistical significance between Low and High groups using distance matrices for beta–diversity Unweighted UniFrac distance. Statistics were calculated using pairwise PERMANOVA with 999 permutations. * *p* < 0.05. Ellipses represent 95% confidence interval for each group.

**Figure 4 jpm-11-01246-f004:**
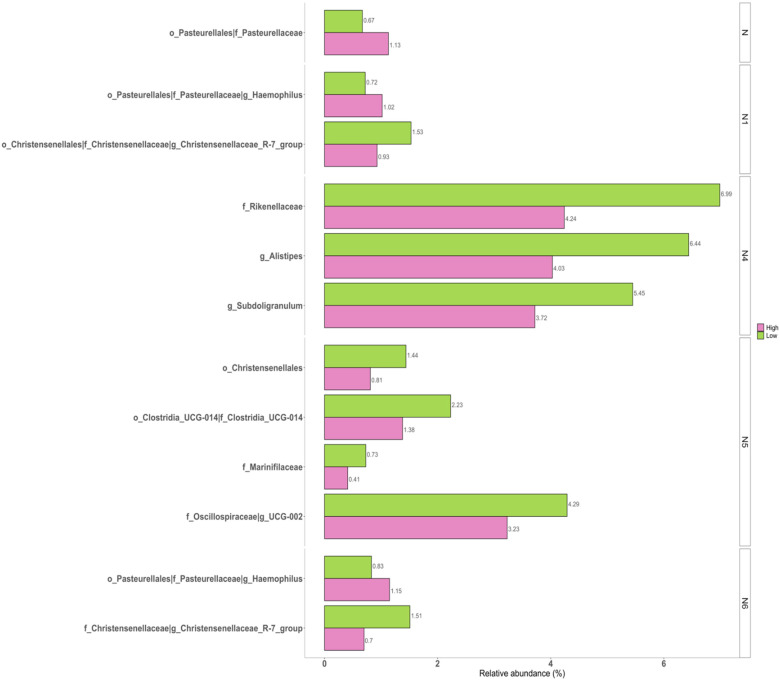
Bar plots for relative abundance of the significant taxa in low and high groups of neuroticism sub-facets. The *x*-axis shows the mean proportion of the significantly different taxa between the low and high groups. The taxa showed the same relative abundance at every level.

**Figure 5 jpm-11-01246-f005:**
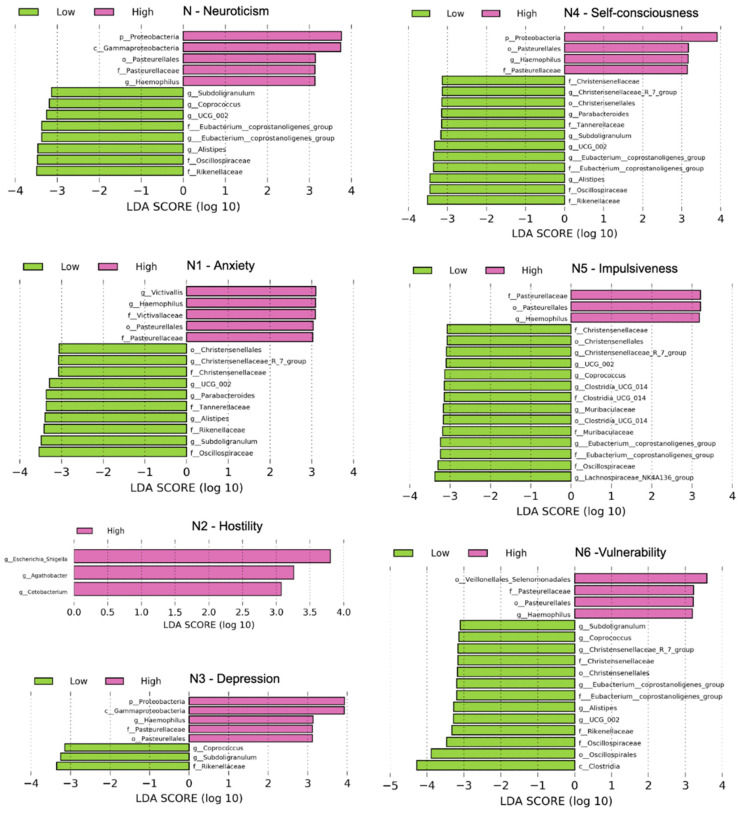
Differentially abundant bacterial taxa in fecal samples from the low and high groups of neuroticism scores. A forest plot showing taxa that were significantly differentially abundant between low (green) and high (pink) as determined using the Kruskal–Wallis test. LDA score (effect size) indicates significant differences in bacterial taxa (LDA score > 3.0); alpha value *p* < 0.05.

**Table 1 jpm-11-01246-t001:** Baseline characteristics of study participants according to neuroticism score.

	Total		Low		High		*p*-Value ^1^
Neuroticism							
No. of subjects	398		205		193		
Age	43.4	(8.2)	44.2	(7.9)	42.6	(8.5)	0.050
BMI	23.8	(3.3)	23.6	(3.3)	24.0	(3.4)	0.314
N score	126.9	(28.0)	102.2	(11.2)	153.1	(12.1)	
Total energy intake	1471.6	(650.8)	1401.0	(608.9)	1546.8	(686.6)	0.046 *
Carbohydrate	244.6	(112.3)	234.1	(105.3)	255.9	(118.7)	0.084
Protein	50.8	(25.5)	48.8	(24.3)	52.9	(26.6)	0.151
Fat	30.7	(19.8)	28.5	(17.8)	33.1	(21.5)	0.041 *
Fiber	3.8	(2.2)	3.7	(2.1)	4.0	(2.4)	0.354
N1 Anxiety							
No. of subjects	407		218		189		
Age	43.5	(8.2)	44.7	(8.3)	42.2	(8.0)	0.002 **
BMI	23.6	(3.2)	23.7	(3.2)	23.5	(3.2)	0.422
N score	22.9	(6.5)	17.3	(2.5)	29.3	(2.7)	
Total energy intake	1483.3	(625.1)	1433.2	(581.0)	1537.2	(667.0)	0.134
Carbohydrate	249.3	(108.5)	242.0	(103.3)	257.3	(113.6)	0.204
Protein	50.3	(23.8)	49.0	(21.8)	51.7	(25.7)	0.292
Fat	30.1	(19.0)	28.5	(16.5)	31.7	(21.3)	0.125
Fiber	3.8	(2.2)	3.9	(2.4)	3.8	(2.0)	0.706
N2 Hostility							
No. of subjects	382		220		162		
Age	43.8	(8.1)	43.9	(8.1)	43.6	(8.2)	0.755
BMI	23.7	(3.3)	23.5	(3.1)	23.9	(3.5)	0.331
N score	19.2	(6.0)	14.5	(2.2)	25.7	(2.5)	
Total energy intake	1452.2	(617.8)	1412.9	(587.1)	1509.6	(658.2)	0.186
Carbohydrate	245.0	(109.1)	239.4	(102.3)	253.2	(118.3)	0.291
Protein	49.7	(24.0)	48.4	(23.3)	51.6	(24.9)	0.260
Fat	28.8	(17.8)	27.4	(17.2)	30.8	(18.6)	0.106
Fiber	3.9	(2.4)	3.8	(2.2)	4.1	(2.5)	0.355
N3 Depression							
No. of subjects	413		236		177		
Age	43.6	(8.2)	43.1	(8.0)	44.3	(8.5)	0.143
BMI	23.9	(3.3)	24.0	(3.3)	23.8	(3.3)	0.595
N score	19.2	(6.5)	14.1	(2.2)	26.0	(3.2)	
Total energy intake	1450.3	(598.6)	1440.0	(549.8)	1463.7	(658.9)	0.729
Carbohydrate	242.8	(106.0)	241.9	(96.5)	244.1	(117.6)	0.851
Protein	49.6	(22.6)	49.7	(22.2)	49.6	(23.1)	0.851
Fat	29.6	(17.9)	28.8	(16.7)	30.7	(19.4)	0.353
Fiber	3.8	(2.0)	3.7	(2.0)	3.8	(2.1)	0.667
N4 Self-consciousness							
No. of subjects	446		252		194		
Age	43.8	(8.4)	43.6	(8.1)	43.9	(8.7)	0.710
BMI	23.7	(3.3)	23.6	(3.2)	23.7	(3.3)	0.866
N score	23.5	(5.3)	19.3	(2.3)	29.0	(1.9)	
Total energy intake	1438.2	(602.6)	1396.9	(584.4)	1492.8	(623.7)	0.143
Carbohydrate	239.0	(103.8)	230.1	(96.6)	250.8	(111.8)	0.069
Protein	49.9	(24.3)	49.4	(25.2)	50.6	(23.2)	0.658
Fat	29.8	(19.8)	29.5	(20.1)	30.3	(19.6)	0.693
Fiber	3.7	(2.1)	3.6	(2.0)	3.8	(2.3)	0.369
N5 Impulsiveness							
No. of subjects	407		228		179		
Age	44.0	(8.2)	45.5	(7.5)	42.1	(8.7)	<0.001 **
BMI	23.6	(3.1)	22.9	(2.9)	24.4	(3.3)	<0.001 **
N score	20.7	(5.5)	16.2	(1.9)	26.5	(2.3)	
Total energy intake	1516.4	(652.1)	1388.1	(538.3)	1689.3	(747.5)	<0.001 **
Carbohydrate	254.6	(112.2)	240.3	(97.2)	274.0	(127.5)	0.009 **
Protein	51.6	(25.2)	45.9	(19.6)	59.2	(29.6)	<0.001 **
Fat	30.7	(20.6)	25.4	(14.6)	37.9	(25.1)	<0.001 **
Fiber	4.0	(2.4)	3.8	(2.1)	4.2	(2.7)	0.150
N6 Vulnerability							
No. of subjects	407		235		172		
Age	44.0	(8.0)	44.8	(7.5)	43.0	(8.5)	0.025 *
BMI	23.8	(3.2)	23.7	(3.2)	23.9	(3.2)	0.698
N score	19.5	(5.4)	15.4	(2.2)	25.1	(2.8)	
Total energy intake	1467.9	(614.8)	1400.4	(569.1)	1561.0	(663.8)	0.024 *
Carbohydrate	245.2	(107.7)	237.0	(101.3)	256.6	(115.4)	0.115
Protein	50.4	(23.1)	48.2	(21.9)	53.5	(24.5)	0.049 *
Fat	30.1	(18.2)	27.3	(15.7)	34.0	(20.6)	0.002 **
Fiber	3.9	(2.3)	3.8	(2.2)	3.9	(2.3)	0.755

^1^ *p*-value for difference between low and high groups by t test for continuous variables. * *p* < 0.05, ** *p* < 0.01.

**Table 2 jpm-11-01246-t002:** Detection of differentially abundant taxa between the two groups for neuroticism scores and coefficients from the generalized linear model using MaAsLin on pairwise testing between the two groups.

Taxa		W ^1^ (Coefficients ^2^) from the Pairwise Groups						
		N	N1	N2	N3	N4	N5	N6
family	p__Bacteroidota; c__Bacteroidia; o__Bacteroidales; f__Marinifilaceae						76 (−0.905 ^4^)	
family	p__Bacteroidota; c__Bacteroidia; o__Bacteroidales; f__Rikenellaceae					71 (−0.896 ^4^)		
genus	p__Bacteroidota; c__Bacteroidia; o__Bacteroidales; f__Rikenellaceae; g__Alistipes					204 (−0.844 ^4^)		
order	p__Firmicutes; c__Clostridia; o__Christensenellales		41 (−1.251 ^4^)				39 (−0.584)	
family	p__Firmicutes; c__Clostridia; o__Christensenellales; f__Christensenellaceae		71 (−1.251 ^4^)					68 (−1.055 ^4^)
genus	p__Firmicutes; c__Clostridia; o__Christensenellales; f__Christensenellaceae; g__Christensenellaceae_R.7_group		221 (−1.249 ^4^)					215 (−0.981 ^4^)
order	p__Firmicutes; c__Clostridia; o__Clostridia__UCG.014						44 (−0.803 ^3^)	
family	p__Firmicutes; c__Clostridia; o__Clostridia__UCG.014; f__Clostridia_UCG.014						73 (−0.803 ^3^)	
genus	p__Firmicutes; c__Clostridia; o__Oscillospirales; f__Oscillospiraceae; g__UCG.002						216 (−0.736 ^3^)	
genus	p__Firmicutes; c__Clostridia; o__Oscillospirales; f__Ruminococcaceae; g__Subdoligranulum					215 (−0.766 ^4^)		
order	p__Proteobacteria; c__Gammaproteobacteria; o__Pasteurellales	43 (1.411 ^4^)	43 (1.112 ^4^)					42 (1.062 ^4^)
family	p__Proteobacteria; c__Gammaproteobacteria; o__Pasteurellales; f__Pasteurellaceae	79 (1.411 ^4^)	77 (1.111 ^4^)					77 (1.061 ^4^)
genus	p__Proteobacteria; c__Gammaproteobacteria; o__Pasteurellales; f__Pasteurellaceae; g__Haemophilus		231 (1.113 ^4^)					218 (1.061 ^4^)

Adjusted for age, sex, and BMI. N # of order 48, # of family. N1 # of order 49, # of family: 87, # of genera 250. N4 # of family: 86, # of genera 251. N5 # of order 50, # of family: 89, # of genera 254. N6 # of order 48, # of family: 85, # of genera 245. ^1^ W = X for taxon k, then H0k is rejected X times. The W statistic for the significantly different taxa relative to more than 70% other taxa in each taxa level is shown in bold. p_ = phlyum; c_ = class; o_ = order; f_ = family; g_ = genus. ^2^ The coefficients from the generalized linear model using MaAsLin on pairwise testing between the two groups. ^3^ *p* < 0.05. ^4^ *p* < 0.01.

## Data Availability

The data presented in this study are available from the corresponding author upon reasonable request.

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
