# Peer review of "Correlation between Gut Microbiota and Six Facets of Neuroticism in Korean Adults"

_jpm, 2021, doi:10.3390/jpm11121246_

Round 1
Reviewer 1 Report
- Line201: Figure 2A was mentioned. However, there was no Figure 2A in Figure 2. It should be corrected.
- Line 204: Beta diversity analysis were described, but there were no figures about beta diversity analysis in figures.
- The data about functionally predictions can be added in the supplement materials.
- In Table 1, N1-N3 was anxiety. In Line 130-132, N1-N3 were anxiety, angry hostility, and depression respectively. They were not identical.
Author Response
Reviewer 1.
- Line201: Figure 2A was mentioned. However, there was no Figure 2A in Figure 2. It should be corrected.
We are sorry for this confusion. According to the reviewer’s comment we corrected the text Figure 2A to Figure 2. (Line 223)
- Line 204: Beta diversity analysis were described, but there were no figures about beta diversity analysis in figures.
Beta diversity analysis represented as a table (Supplementary Table S1). We represented PCoA plots significant results only (figure 3). We corrected text referring Figure 3 and added Supplementary table S1.
(Line 226) Beta diversity analysis indicates the extent of similarities and differences among microbial communities. To quantify beta diversity, both phylogenetic and non-phylogenetic methods were used with unweighted and weighted UniFrac and Jaccard distances and Bray–Curtis dissimilarity, respectively (Figure 3). For the neuroticism domain, we confirmed significant differences between the low and high groups in unweighted Unifrac distance (p<0.05, PERMANOVA), similar to the results of our previous study [13]. For facets of neuroticism, there were significant differences in unweighted distances between the high and low groups of the N1 anxiety, N4 self-consciousness, N5 impulsiveness, and N6 vulnerability facets, respectively. For N2 and N3, we could not find statistical differences in the structure of the gut microbial community between the low and high groups (Supplementary Table S1).
- The data about functionally predictions can be added in the supplement materials.
We agreed to reviewer’s opinion. However, STAMP program which we used for visualizing PICRUST2 results represents only significant results. Therefore, we are sorry that we could not add supplement materials of functionally predictions. We are trying to find an alternative visualization program that can be used in further research.
- In Table 1, N1-N3 was anxiety. In Line 130-132, N1-N3 were anxiety, angry hostility, and depression respectively. They were not identical.
According to the reviewer’s comment we corrected words in the table.
Reviewer 2 Report
-In this manuscript, the authors researched the Correlation between gut microbiota and six facets of neuroticism in Korean adults. The manuscript was very exciting to be read and presents interesting data sets as obtained from Korean adults’ analysis. This paper can be interesting but the Authors should explain in more points with gut microbiota. The manuscript possesses novelty and therefore it has potential to interest the readers from gut microbiota. However, the following comments must be answered by the authors prior to publication.
Minner Comments:
Q1: In page 2, line 57-78, modification is here required, divide this paragraph as two parts.
Q2: In figure 1, modification is required. Adjust the empty space and make compact size of figure 1?
Q3: Significance of human gut microbiome has documented. Write few more lines about gut microbiome in introduction section.
Q4: In Figure 2, text size like low and high is very poor in all figures. It should be improve in x and y-axis.
Author Response
Q1: In page 2, line 57-78, modification is here required, divide this paragraph as two parts.
Q3: Significance of human gut microbiome has documented. Write few more lines about gut microbiome in introduction section.
According to the reviewer’s comment Q1 and Q3, We added more detailed information about gut microbiome and divided into two paragraphs to improve readibiliity.
(Line 57) Recently, the importance of the gut microbiome in human health has been found the spotlight [11]. Since the gut microbiota play a central role in the gut–brain axis that regulates mood and behavior, they can affect various aspects of normal psychology, such as emotion, cognition, stress management, and social behavior in addition to physical health [6, 12]. The gut microbiota are also correlated with the predisposition of personality and mental disorders [13, 14]. Dysbiosis in the gut microbiota may increase pro-inflammatory communication, which in turn increases intestinal permeability, which can lead to an inflammatory response to stress systems of the brain either directly or vagus/visceral afferent [11, 12]. Pro-inflammatory communication has been shown to impair negative feedback within the hypothalamic–pituitary–adrenal (HPA) axis and induce hypercortisolemia [11]. Elevated cortisol levels and inflammatory markers are reported to be associated with anxiety and depressive disorders [12]. This bidirectional communication implicates that increased cortisol delivered to the body can affect immune function, intestinal permeability, and gut microbiota [12].
In a previous study, we have reported the correlations between that gut microbiome and personality traits. Highly neurotic individuals were likely to show a high abundance of Gammaproteobacteria. However, the subscales of neuroticism were not considered in that study. Recent research has suggested that compositions of the human microbiome are linked to stress, depression, and broad personality traits [14-17]. Chronic stress can increase the risk of developing many psychiatric disorders such as anxiety, depression, and post-traumatic stress disorder [18, 19]. Although the mechanisms underlying vulnerability to stress remain unclear, mounting evidence suggests that increased central inflammatory processes might be involved [20-22]. Indeed, the gut microbiome contributes to the depression-like behaviors and inflammatory processes in the ventral hippocampus of stress vulnerable rats [15]. Although the Revised NEO Personality Inventory (NEO-PI-R) contains the six facets that are the subscales of neuroticism, there are no reported studies on the correlation between these neuroticism facets and the gut metagenome. Neuroticism is hierarchically defined by the specific facets, which can provide a more in-depth description of the correlation between personality and gut microbiota. Moreover, as a more circumscribed facet, the trait of neuroticism measures a narrower phenotype, which can increase the statistical power by reducing phenotyping variability [23].
Q2: In figure 1, modification is required. Adjust the empty space and make compact size of figure 1?
According to the reviewer’s comment, we resized figure 1 to make it compact and fit.
